

# Mercuric pollution of surface water, superficial sediments, Nile tilapia (*Oreochromis nilotica* Linnaeus 1758 [Cichlidae]) and yams (*Dioscorea alata*) in auriferous areas of Namukombe stream, Syanyonja, Busia, Uganda

Timothy Omara[1,2,3,*], Shakilah Karungi[4,*], Raymond Kalukusu[2,5], Brenda Victoria Nakabuye[5,6], Sarah Kagoya[2,7] and Bashir Musau[2,5]

[1] Department of Chemistry and Biochemistry, School of Biological and Physical Sciences, Moi University, Uasin Gishu County, Kesses, Eldoret, Kenya

[2] Department of Chemistry, Faculty of Science, Kyambogo University, Kyambogo, Kampala, Uganda

[3] Department of Quality Control and Quality Assurance, Product Development Directory, AgroWays Uganda Limited, Kyabazinga way, Jinja, Uganda

[4] Department of Mining and Water Resources Engineering, Faculty of Engineering, Busitema University, Busitema, Tororo, Uganda

[5] Department of Quality Control and Quality Assurance, Leading Distillers Uganda Limited, Kampala, Uganda

[6] Department of Food Processing Technology, Faculty of Science, Kyambogo University, Kyambogo, Kampala, Uganda

[7] Department of Quality Control and Quality Assurance, Product Development Directory, Sweets and Confectionaries Section, Kakira Sugar Limited, Jinja, Uganda

[*] These authors contributed equally to this work.

Corresponding author
Timothy Omara,
prof.timo2018@gmail.com

## ABSTRACT

The mercury content and the contamination characteristics of water, sediments, edible muscles of a non-piscivorous fish (*Oreochromis nilotica* Linnaeus 1758 [Cichlidae]) and yams (*Dioscorea alata*) from Namukombe stream in Busia gold district of Uganda were evaluated. Human health risk assessment from consumption of contaminated fish and yams as well as contact with contaminated sediments from the stream were performed. Forty-eight (48) samples of water ($n = 12$), sediments ($n = 12$), fish ($n = 12$) and yams ($n = 12$) were taken at intervals of 10 m from three gold recovery sites located at up, middle and down sluices of the stream and analyzed for total mercury (THg) using US EPA method 1631. Results (presented as means $\pm$ standard deviations) showed that water in the stream is polluted with mercury in the range of < detection limit to 1.21 $\pm$ 0.040 mg/L while sediments contain mean THg from < detection limit to 0.14 $\pm$ 0.040 $ugg^{-1}$. Mean THg content of the edible muscles of *O. nilotica* ranged from < detection limit to 0.11 $\pm$ 0.014 $ugg^{-1}$ while *D. alata* contained from < detection limit to 0.30 $\pm$ 0.173 $ugg^{-1}$ mean THg. The estimated daily intake ranged from 0.0049 $ugg^{-1}day^{-1}$ to 0.0183 $ugg^{-1}day^{-1}$ and 0.0200 $ugg^{-1}day^{-1}$ to 0.0730 $ugg^{-1}day^{-1}$ for fish consumed by adults and children respectively. The corresponding health risk indices ranged from 0.0123 to 0.0458 and 0.0500 to 0.1830. Estimated daily intake was from 0.0042 $ugg^{-1}day^{-1}$ to 0.1279 $ugg^{-1}day^{-1}$ and 0.0130 $ugg^{-1}day^{-1}$ to 0.3940 $ugg^{-1}day^{-1}$ for *D. alata* consumed by adults and children respectively. The

health risk indices recorded were from 0.011 to 0.320 and 0.033 to 0.985 for adults and children respectively. The mean THg content of the sediments, edible muscles of *O. nilotica* and *D. alata* were within acceptable WHO/US EPA limits. About 91.7% of the water samples had mean THg above US EPA maximum permissible limit for mercury in drinking water. Consumption of *D. alata* grown within 5 m radius up sluice of Namukombe stream may pose deleterious health risks as reflected by the health risk index of 0.985 being very close to one. From the pollution and risk assessments, mercury use should be delimited in Syanyonja artisanal gold mining areas. A solution to abolish mercury-based gold mining in the area needs to be sought as soon as possible to avert the accentuating health, economic and ecological disaster arising from the continuous discharge of mercury into the surrounding areas. Other mercury-free gold recovering methods such as use of borax, sluice boxes and direct panning should be encouraged. Waste management system for contaminated wastewater, used mercury bottles and tailings should be centralized.

# INTRODUCTION

For centuries, the most precious and enigmatic metals (mercury and gold) have been linked both chemically and in utilization by man due to their demonstrated propensity to produce an amalgam when mixed (*Legg, Ouboter & Wright, 2015*). Mercury (Hg) on the other hand received further recognition for its toxicity when used in gold recovery since the Roman times (*Legg, Ouboter & Wright, 2015*). As such, increase in the price of gold, lack of regulation, chronic unemployment and increasing poverty have led to an explosive growth in global artisanal and small-scale gold mining (ASGM) in the last century. The activity is emerging as a significant socio-economic sector with the result that it has become a major source of revenue for more than 100 million people in over 80 countries of Sub-Saharan Africa, Asia, Oceania, Central and South America (*United Nations Environment Programme, 2013*). This is because it involves low or no capital investment and mechanization, and in most cases the sector is not taxed by national revenue authorities.

Globally, ASGM is the single largest demand for mercury and thus the largest source of intentional pollution of air and water combined (*United Nations Environment Programme, 2012*). In Uganda, ASGM was conservatively estimated to employ over 60,000 vulnerable people as of 2016 (*Africa Centre for Energy and Mineral Policy, 2017*). ASGM is being practiced in the gold districts of Mubende, Namayingo, Bugiri, Mayuge, Busia, Buhweju, Bushenyi, Ibanda, Kanungu, Moroto, Abim, Kaabong, Nakapiripirit and Amudat (*Africa Centre for Energy and Mineral Policy, 2017*). Gold in Uganda was first reported in West Nile in 1915 though no mining commenced until 1933. In Busia, gold was reported in 1932 by Davis in Osipiri area (Neoarchean Busia-Kakamega granite greenstone belt) (*Combe, 1933*) which registered intermittent ASGM on vein and alluvial deposits in Tiira, Makina,

Amonikakine and Osapiri villages up-to-date (*Africa Centre for Energy and Mineral Policy, 2017*). The belt is an extension of the northern Tanzanian greenstone belt where gold occurs in association with quartz veins. Thus, mercury-based ASGM was introduced in the 1990s by Ugandans who remigrated into Uganda from the Tanzanian greenstone belt (*Africa Centre for Energy and Mineral Policy, 2017*). Unlike the small mechanized processing centres in Tanzania, miners in Busia use manual methods characterized by pounding of auriferous materials, hand amalgamation in basins (panning), open air burning over fires and haphazard discharge of tailings into waterways. Panning is a simple process which separate particles of greater specific gravity (such as gold) from soils, gravels and sediments using water in a dish called a pan.

Despite the fact that rudimentary tools are employed, ASGM in Busia has continued, an indication that the business is recovering sizeable quantities of gold nuggets (*Africa Centre for Energy and Mineral Policy, 2017*). More so, the price of gold now exceeds US $1,600 per ounce (from less than US $500 in the 1980s), causing ASGM to rise along with its elemental mercury pollution (*Armah, 2013*). A report from the Bank of Uganda cited gold as the second most important export of Uganda after coffee with a worth of US $ 35.73 million in 2015 and US $ 339.54 million in 2016 (*The New Vision, 2018*).

In Busia and Bugiri districts, close to 1,000 gold miners engage in mercury-based ASGM with 150 kg of mercury per annum reported to end up in the downstream areas (*United Nations Environment Programme, 2012*). Approximately, 45 kg of this mercury get discharged with tailings into small rivers and streams in Busia during gravity concentration of auriferous materials (panning) (*United Nations Environment Programme, 2012*). Miners are thus exposed to elemental mercury intoxication (*Veiga & Baker, 2004*) as well as cyanide used to extract gold from tailings. Similarly, miners handling mercury used for amalgamation and those living in close proximity to uncontrolled mercury-based gold recovering sites risk coming in contact with mercury through dermal adsorption during amalgamation, inhalation of mercury vapor, drinking mercury-polluted water and consumption of food such as fish, yams, maize, millet and potatoes grown in mercury-contaminated water and soils.

The population of Tiira village in Busia who participated in mercury-based ASGM for the past decades have been diagnosed with severe health complications including paralysis (*Daily Monitor, 2019*). The miners do not use any form of personal protective equipment while using mercury (*Daily Monitor, 2019*) which is contrary to international guidelines on mercury handling (*ACCC, 2018*). It is reported that mercury is poorly recovered in ASGM processes and the emission factor can be as high as 15 g of mercury per gram of gold recovered (*Streets et al., 2005*). Thus, ASGM is inextricably linked with human health, social, environmental and economic problems (*Daily Monitor, 2019*).

Elemental mercury intoxication causes irreversible neurological, kidney and autoimmune impairment accompanied by respiratory tract irritation, chemical pneumonitis, pulmonary oedema, chest tightness, respiratory distress (*Agency for Toxic Substances and Disease Registry, 2014*), respiratory failure and subsequently death (*Landrigan & Etzel, 2013*). Systemic absorption of elemental mercury causes nausea, vomiting, headache, fever, chills, abdominal cramps and diarrhea. Chronic, lower level

exposure to elemental mercury induces gingivostomatitis, photophobia, tremors and neuropsychiatric symptoms (*World Health Organization, 2003*). Elemental and inorganic mercury toxicity in children may be witnessed in oedematous, painful, red, desquamating fingers and toes (acrodynia) as well as hypertension (*Bose-O'Reilly et al., 2010*).

Mercury used in recovering gold by artisanal miners may get transmogrified along the food chain into methyl mercury which is the most toxic organic form of mercury that can bioaccumulate in the food chain (*Bose-O'Reilly et al., 2010*; *Spiegel, 2009*). Methyl mercury because of its lipid-solubility readily enter the bloodstream via the digestive system (*World Health Organization, 1990*). On crossing the blood–brain barrier, methyl mercury accumulates in the spinal cord, triggering headache, ataxia, dysarthria, visual field constriction, blindness, hearing impairment, psychiatric disturbance, muscle tremor, movement disorders, paralysis and death (*Gibb & O'Leary, 2014*).

Uganda National Environmental Management Authority reported in 2017 that the Ugandan ASGM sector contributes an estimated annual mercury input of 18.495 tons, of which 12.136 tons are released in the air, 3.333 tons are released in water and 3.027 tons are disposed on land (*National Environment Management Authority, 2012*). As reported in other ASGM areas globally, wastes containing residual mercury from ASGM are often discharged in a perverted fashion into fragile ecosystems (*Pfeiffer & De Larceda, 1988*; *Meech, Veiga & Tromans, 1998*), causing prodigious pollution of water, sediments, biota, soil and air (*Santos-Frances et al., 2011*; *Ogola, Mitullah & Omulo, 2002*; *Garcia-Sanchez et al., 2016*). This is disastrous for the case of Syanyonja and Busia as a whole since the mercury can get entrained in sediments which are carried to "the life artery of East African countries" (Lake Victoria), located 30 km downstream (*United Nations Environment Programme, 2012*).

This study provides the first ever comprehensive assessment of the mercury contamination of water, sediment, fish and yams from Namukombe stream in Syanyonja village, Busia gold district, Uganda and create a paradigm for future studies aimed at developing strategies for reducing mercury pollution from ASGM in Busia. The results of this study is also a resource to Uganda National Environmental Management Authority and Ministry of Energy and Mineral Development.

## MATERIALS AND METHODS

### Description of study area

Busia gold district is located between 33°05′E 00°10′N and 34°01′E 00°35′N. It covers 730.9 square kilometres of land and is bordered by Tororo district to the North, Kenya to the East, Lake Victoria to the South, Namayingo district to the Southwest and Bugiri district to the West. Available land area is 648.95 square kilometres while open water and swamps cover about 36.88 square kilometres (*ACCC, 2018*). It has 10 sub counties, 58 parishes and 609 villages characterized by undulating plain topography with an altitude of 1128 m above sea level at Nebolola hills in Lumino sub county. Low lying areas (valleys) lie at altitudes of 1,000 m above sea level with River Malaba valley to the North and River Lumboka to the West (Fig. 1). The landscape is comprised of a mixture of auriferous hard rock (70%) and 30% alluvial soil (*ACCC, 2018*).

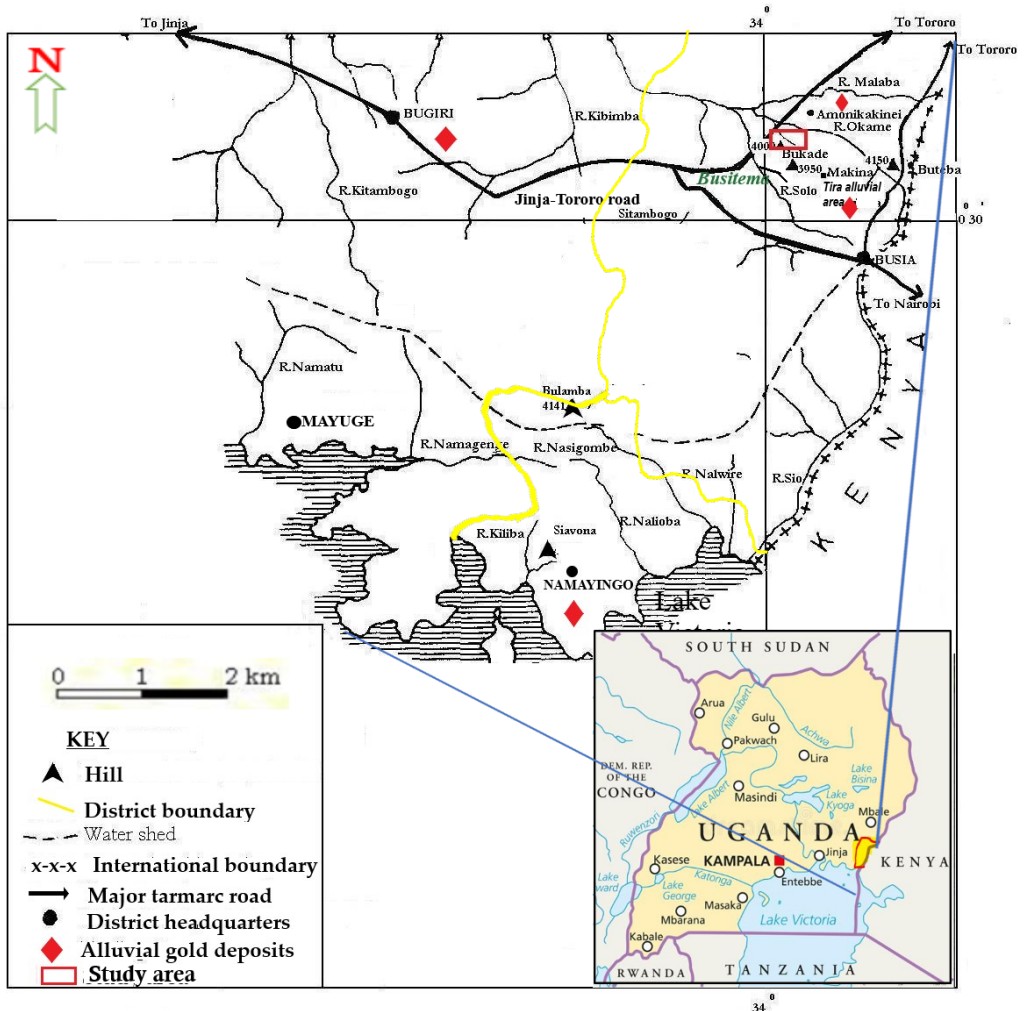

**Figure 1** **Map of Busia gold district showing the location of Namukombe stream.** Inset is the location of Busia gold district on the map of Uganda (adapted from *Mbonimpa, 2005*).

Busia experience a bimodal rainfall pattern with an average annual rainfall of 1,727 mm. The first dry rainy season (short rains) start from March to May, followed by a longer rainy season between August to November. This climatological characteristic facilitates the leaching of poorly managed mercury wastes from ASGM sites into water bodies in the surrounding areas (*ACCC, 2018*).

Busia is characterized by two distinct hydrographical networks (Fig. 1). There is a northern network with River Malaba, River Kibimba and River Kitambogo draining into Lake Kyoga and a southern network with rivers: Sio, Nasigombe, Nalioba, Namagenge, Nalwire and Namatu draining into Lake Victoria (*Mroz et al., 1991*). All the streams in the study area (such as Solo, Namukombe, Dad, Okame, Nakola, Tira and Osapiri) are tributaries of River Malaba.

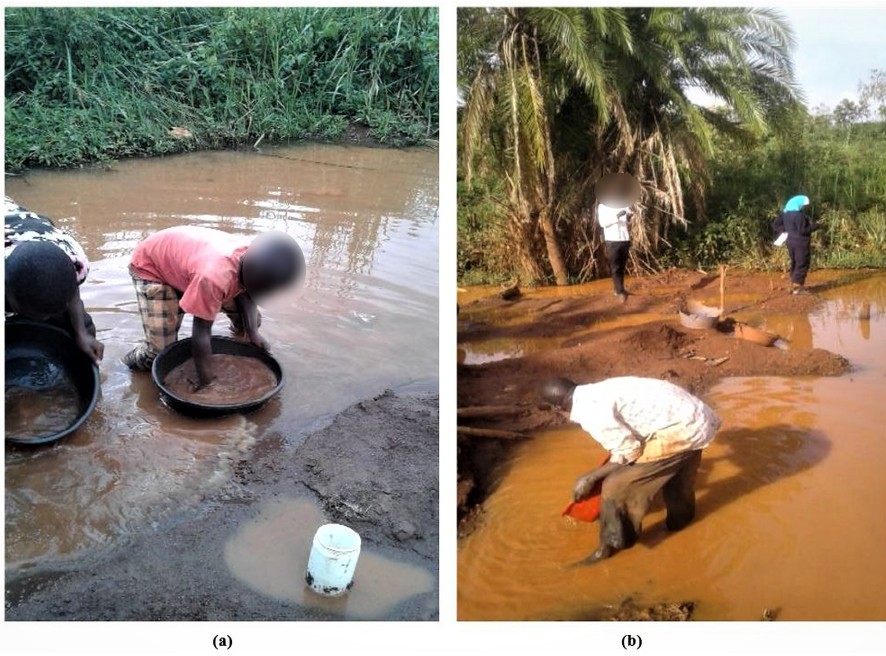

**Figure 2** **Panning of alluvium in Namukombe stream.** (A) at the first gold recovery site (up sluice) by school dropouts (children); (B) at the third gold recovery site (down sluice) by a man (Photo credit: Shakilah Karungi).

The current study was conducted in Namukombe stream in Syanyonja village, Samia Bugwe constituency, Busitema sub county ($00^0 34' 21''$N $34^0 03' 00''$E), Busia gold district, South East of the Republic of Uganda. Syanyonja village lies in the coordinates UTM 60648 36N617118 (*Rwabwoogo, 1998*). Namukombe stream, a tributary of River Malaba, is one of the major water bodies in Syanyonja village where mercury-based ASGM activities and fishing are done (Fig. 2). Water for drinking and domestic use are collected from the stream because the community have few boreholes and permanent wells as alternative water sources. Growing of food crops (millet, cassava, rice, sweet potatoes, yams, simsim, sorghum, cow peas and bananas) (*Mbonimpa, 2005*) are done along the banks of the stream. The most important activities in Busia are fishing and the cross-border trade (*COWI, 2016*). A significant portion of the population engages in artisanal mining and panning of soil and stream sediments for gold (*Mbonimpa, 2005*).

Thus, the selection of the study area was based on the fact that the stream receives drainage from three artisanal mercury-based gold recovering sites (1, 2 and 3; Fig. 3) located approximately 800 m from each other and designated as up sluice, middle sluice and down sluice respectively (Fig. 4). Wastewater from these gold recovery sites are discharged directly into the stream.

## Ethical approval

This research required no ethical approval. However, field experiments were approved by Department of Mining and Water Resources Engineering, Faculty of Engineering,
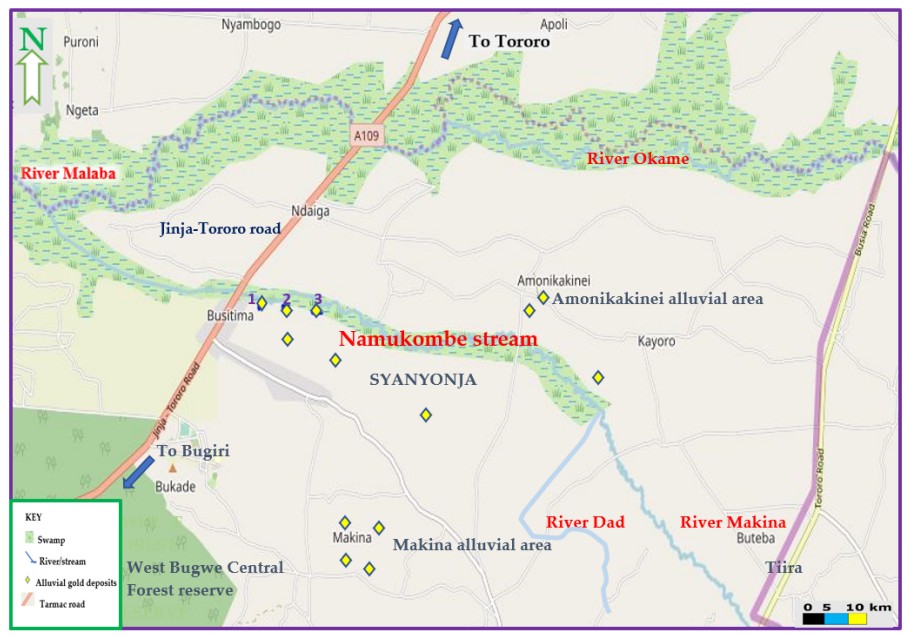

**Figure 3** **Map of Namukombe stream showing the location of the sampled sluices.** 1, 2 and 3 are the gold recovering sites located along the banks of the stream approximately 800 m from each other. Map data ⓒ2019 OpenStreetMap contributors.

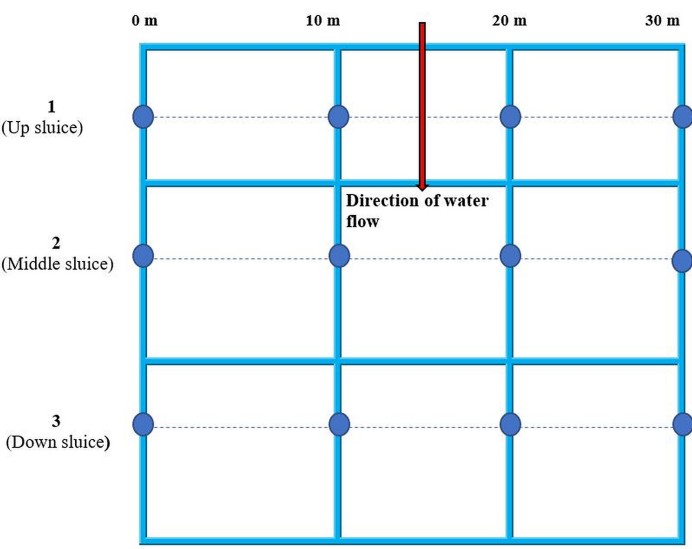

**Figure 4** **Sampling plan for water, sediments, fish and yams from Namukombe stream.** Yam and fish samples were taken from within a maximum of 5 m radius from the sampling points indicated in the plan.

Busitema University, Tororo, Busitema, Uganda for Shakilah Karungi (Approval No. **BU/UG/2013/95**)

## Collection and preparation of samples

Water and sediments were selected to determine the concentration of mercury being released into the stream with wastewater. Fish and yams were selected in this study because they constitute the daily diet of the local community. Some miners eat smoked fish without boiling.

Samples were obtained in triplicate from up, middle and down sluices at 0 m, 10 m, 20 m and 30 m. Fish and yam samples were obtained at most from within a radius of 5 m from the sampling points indicated in Fig. 4.

Water samples were collected manually in 500 ml Teflon plastic bottles. The bottles, previously cleaned by washing in non-ionic detergent, were rinsed with tap water, soaked in 10% nitric acid for 72 h and finally rinsed with deionized water before use. Each bottle was rinsed 3 times with stream water at each sampling point, filled 20–25 cm below the water surface and capped with airtight stoppers while still under water. The samples were filtered through predried polyvinylidene difluoride membrane filters (Millipore, 0.45 $\mu$m pore diameter) and preserved with 5 ml of concentrated nitric acid to achieve pH <2.0 (to minimize precipitation and sorption losses to the bottle walls).

Superficial sediment samples were obtained using a grab sampler at 0–5 cm in accordance with the United Nations Environment Programme's reference method for sediment pollution studies (UNEP method number RSRM 20) (*UNEP, 1992*) to preclude contamination and address geographic differences. Sediment samples collected were put in clean presterilized plastic polypropylene bottles, tightly sealed and labelled. The samples were oven-dried at 80–95 °C for 16 h until a constant weight was attained. They were subsequently crushed in a stone mortar and sieved through a 0.63 $\mu$m nylon mesh sieve. Particle size fraction of <63 $\mu$m for analysis is recommended as it is the most near to be taken as equivalent to materials carried in suspension (*Salomons & Forstner, 1984*). The powdered sediment samples were preserved at 4 °C on an ice block.

Fish samples (5.5 to 8.0 cm fork length) were caught using hooks fed with earthworms as baits. The samples were preliminarily prepared for analysis as described by *Omara et al. (2018)*.

Yam tubers obtained were manually peeled using a stainless-steel knife and cut into small pieces. Aliquots (3.0 $\pm$ 0.1 g) were weighed and ashed in a furnace at 550 °C for 5 h.

## Analytical reagents and apparatus used

Standard mercury solution was purchased from Sigma Aldrich (Buchs, Switzerland). The other reagents, except certified reference materials, were obtained from Merck (Darmstadt, Germany). Water used as solvent was deionized water.

The assortment of volumetric ware used were soaked overnight in 10% (v/v) nitric acid solution, rinsed with deionized water (*Khansari, Ghazi-Khansari & Abdollahi, 2005*) and oven dried prior to analysis.

Mettler PM200 digital analytical balance (Marshall Scientific, Hampton, NH, USA) was used for all analytical weighing. Hanna 211 digital microprocessor-based bench top

pH/mV/°C meter (Hanna instruments, Italy) calibrated using pH 4.01, 7.01, 10 buffers was used for all pH measurements.

## Spectroanalytical procedure
### Water samples
The concentration of total dissolved mercury (THg) in water samples (5 ml aliquots) were determined using cold vapour atomic fluorescence spectrometer (CV-AFS Brooks Rand® Model III; Brooks Rand Instruments, Seattle, WA, USA) in accordance with US EPA method 1631 (*US EPA, 2001*), modified for measurement of low-level mercury in water (*Heimbürger et al., 2015*).

Briefly, 50 ml of the samples were subjected to bromination using 0.2 N bromine monochloride in a ratio of 1: 2 (v/v) to break down any organomercury compounds in the samples to mercury (II) ions. The samples were kept in an oven at 50 °C for 12 h. Prior to analysis, excess bromine (as shown by presence of yellow tints) in the samples was neutralized with 10% hydroxylamine hydrochloride. After 10 min, the samples were then reduced with 5 ml of stannous chloride. Measured 5 ml aliquots of the resultant solutions were purged with mercury-free nitrogen gas for 20 min and trapped on columns packed with gold coated sand. The gold trap was heated, and the desorbed mercury detected with Cold-Vapor Atomic Fluorescence Spectrophotometer following manufacturer's conditions with a practical detection limit of 0.001 mg/L.

## Sediment, fish and yam samples
Digestion of sediment, fish and yam samples were performed using a procedure modified from US EPA method 1631 Appendix for digested (oxidized) solid samples (*US EPA, 2001*) to allow for THg determination using the manual 245.7 system coupled to a ball flowmeter and a multi-channel pump as per cold vapour atomic fluorescence spectrometer (Brooks Rand® Model III; Brooks Rand Instruments, Seattle, WA, USA) manufacturer's instructions (*Brooks Rand Instruments, 2019*). This system is supplied with Brooks Rand Guru™ software for system control and peak integration.

For each sample, three separate aliquots (0.25 g) were transferred to 50 ml polypropylene test tubes. To the aliquots, 5 ml of 3:1 (w/w) sulphuric acid: nitric acid mixture were added and allowed to stand at room temperature (25 °C) for 2 h. The samples were heated to 80 °C for 40 min and subsequently liquefied. 15 ml of 6 N hydrochloric acid, 3 ml of 0.1 N bromine monochloride solution and 4 ml of deionized water were added. The samples were agitated and then heated to 60 °C for 1 h. Fish samples turned yellow in color, with the resultant solutions remaining clear without any precipitate. Each sample was diluted in a ratio of 1:10 with 2% (w/v) hydrochloric acid. Exactly 0.1 ml of 10% hydroxylamine hydrochloride solution was added to remove excess bromine. The samples (5 ml aliquots) were then analyzed for THg using Brooks Rand® Model III cold vapour atomic fluorescence spectrometer using the manual 245.7 system following the manufacturer's instructions. This semi-automated instrument allowed simultaneous purging of multiple samples while previously loaded amalgamation traps were analyzed with a throughput rate of less than 5 min per sample.

The THg concentrations were reported in mg/L for water or $\mu gg^{-1}$ dry weight for solid matrices for easy comparison with set international compliance limits.

## Analytical quality assurance and quality control

All reagents used in this investigation were of high analytical purity. Quality control was performed with certified reference materials and spiked samples analyzed once for every 10 samples. Certified reference materials for water (NRCCORMS-5; National Research council Canada, Ottawa, Canada), sediment (CRM008-050; Resource Technology Corporation, USA) and fish (SRM 2976; National Institute of Standards and Technology, Gaithersburg, MD, USA) were used to assess recovery which yielded respectively 99.6%, 102.8% and 101% of the certified mercury values (Table S1). Mercury recovery percentages from the spiked samples were 98.8%, 98.2%, 97.9% and 99.2% for water, sediment, fish and yam samples respectively. All samples were analyzed at least in triplicate to obtain a relative uncertainty of less than 5%. In other words, precision in the analysis was in good agreement, better than $\pm 5.0\%$ relative standard deviation.

Method detection limit with reagent blanks was calculated and used as a verification tool. The calculated method detection limits were 0.001 mg/L for water and 0.001 $\mu gg^{-1}$ for the solid matrices. Bottle, analytical, equipment and filtration blanks were determined throughout the analyses, and blank subtractions were used to correct the mercury concentrations obtained.

## Human health risk assessment

Estimated daily intake (EDI) was calculated to averagely estimate the daily mercury loading into the body system of a specified body weight of a consumer (adult/child) through consumption of contaminated fish and yams while the average daily dose ($ADD_{therm}$) was calculated to determine intoxication through skin contact with mercury-contaminated sediments. These provide the relative availability of mercury but does not take into cognizance the possibility of metabolic ejection of mercury. EDI in $\mu gg^{-1}day^{-1}$ was calculated using Eq. (1) previously employed elsewhere (*Omara et al., 2018*) while $ADD_{therm}$ ($\mu gg^{-1}day^{-1}$) was calculated from Eq. (2) (*Nowell et al., 2013*; *Superfund Public Health Evaluation Manual, 1986*; *Ordonez et al., 2011*).

$$EDI = \frac{E_f \times E_d \times F_{ir} \times C_f \times C_{hm}}{W_{ab} \times T_{aet}} \qquad (1)$$

$$ADD_{therm} = \frac{C_{hm} \times S_A \times AF \times E_f \times E_d}{W_{ab} \times T_{aet}} \times 10^{-6} \qquad (2)$$

where $E_f$ = exposure frequency (365 days/year), $E_d$ = exposure duration, the average lifetime (58.65 years for an adult Ugandan) (*Bamuwamye et al., 2017*), $F_{ir}$ is the fresh food ingestion rate (g/person/day) = 48 for fish and 301.0 and 231.5 for yams eaten by adults and children respectively) (*Wang et al., 2005*; *Ge, 1992*), $C_f$ is the conversion factor (0.208) for fresh weight ($F_w$) to dry weight ($D_w$) for fish and 0.085 for yams (considering it as a vegetable) (*Adedokun et al., 2016*), $C_{hm}$ = heavy metal concentration in the foodstuffs ($\mu gg^{-1}$ Fw), $W_{ab}$ = average body weight (considered to be 15 kg for children

(*Ordonez et al., 2011*) and 60 kg for adults (*Ali & Hau, 2001*)), $T_{aet}$ = average exposure time for non-carcinogens (given by the product of $E_d$ and $E_f$) (*Saha & Zaman, 2012*), $S_A$ is the exposed surface area in $cm^2$ = 4,350 for adults and 2,800 for children (*Ordonez et al., 2011*), AF is the skin adherence factor in $mg/cm^2/day$ = 0.7 for adults (*Wojciechowska et al., 2019*) and 0.2 for children (*Ordonez et al., 2011*).

Health risk index, the total risk of a non-carcinogenic element through three exposure pathways was evaluated using Target hazard quotient in accordance with US EPA region III risk-based concentration table (*US EPA, 2009*) used in a preceding study (*Omara et al., 2018*). According to the criterion, target hazard quotient less than unity (1.0) indicate that the exposure is very unlikely to have adverse effects while target hazard quotient greater than 1.0 represent a possibility of non-carcinogenic effects, with its probability increasing as the value of the target hazard quotient increases (*Omara et al., 2018*). Target hazard quotient was calculated as the ratio of EDI or $ADD_{therm}$ to the reference dose ($R_f D$) (Eq. (3)) (*Omara et al., 2018*; *Superfund Public Health Evaluation Manual, 1986*).

$$\text{Target hazard quotient} = \frac{EDI}{R_f D} \quad or \quad \text{Target hazard quotient} = \frac{ADD_{them}}{R_f D} \tag{3}$$

The reference dose of mercury through ingestion in $\mu gg^{-1}day^{-1}$ is $4.0 \times 10^{-1}$[41] while the reference dose for mercury through skin contact is $1.0 \times 10^{-2} \mu gg^{-1}day^{-1}$ (*Wang et al., 2005*).

The reference dose is the maximum daily dose of a metal from a specific exposure pathway, that is believed not to lead to an appreciable risk of deleterious effects to sensitive individuals during a life time (*Qing, Yutong & Shenggao, 2015*). If the EDI is lower than the reference dose, the target hazard quotient is less than 1 and adverse health effects are unlikely to appear, whereas if the EDI exceed the reference dose, target hazard quotient is greater than 1 and adverse health effects are likely to appear (*Nowell et al., 2013*; *Superfund Public Health Evaluation Manual, 1986*). In this study, the target hazard quotient was calculated basing on three pathways i.e., consumption of mercury-contaminated fish and yams and skin contact with mercury (miners come into contact with dredged mercury-contaminated sediments during panning).

The assumptions made during the health risk calculations were that the ingested dose is equal to the dose absorbed into the body (*US EPA, 1989*) and cooking of fish and yams have no effect on the mercury content of the assessed matrices (*Cooper, Doyle & Kipp, 1991*).

## Assessment of bioaccumulation factors

Bioaccumulation factors are multipliers used to estimate concentrations of chemicals that accumulate in tissues through any route of exposure (*US EPA, 2000*). They are often referred to as bioconcentration factors for aquatic invertebrates. The bioconcentration factors and biota to sediment accumulation factor of trace metals from sediment or surface water to animal tissues can be determined for different samples. Thus, bioconcentration factor was determined from the numerical ratio of the concentration of mercury in the whole edible tissues of *O. nilotica* to concentration of mercury in water while biota to

sediment accumulation factor was determined from the ratio of the concentration of mercury in the edible muscles of *O. nilotica* to that of mercury in the corresponding sediment samples (*Benson et al., 2017*).

## Sediment quality assessment

To assess the level of contamination as well as the environmental and health risks that originate the occurence of a heavy metal, indices are used to indicate the enrichment of a given environmental component as compared to its natural concentration. The indices used to describe heavy metal enrichment of sediments include contamination factor, enrichment factor, pollution load index, geoaccumulation index, potential ecological risk and hazard quotient (*Hakanson, 1980*; *Tomlinson et al., 1980*; *Zheng et al., 2015*).

These indices are used as a rule at thumb to subtly assess sediment contamination. However, these do not allow comparison of the degrees of contamination of sediments investigated in different studies. To identify pollution problems, the anthropogenic contributions should be distinguished exclusively from the natural sources. Thus, the degree of mercuric contamination of sediments from Namukombe stream was assessed using contamination factor and geoaccumulation index.

The contamination factor was calculated using Eq. (4) given by *Hakanson (1980)*.

$$\text{Contamination factor} = \frac{C_{hm}}{C_b} \tag{4}$$

where $C_{hm}$ is the priority trace metal concentration in the analyzed sample and $C_b$ is the geochemical background trace metal concentration/preindustrial concentration. A background concentration of 0.25 $\mu g g^{-1}$ of mercury in crustal shale was adopted from *Taylor (1964)*.

Geoaccumulation index ($I_{geo}$) for the sediments from the different sluices were obtained from computations utilizing **Eq. (5)** suggested by *Müller (1981)*.

$$I_{geo} = Log_2(\frac{C_n}{1.5B_n}) \tag{5}$$

where $C_n$ is the concentration of the trace metal ($n$) in the sampled and analyzed sediment, $B_n$ is the background concentration of the same metal and 1.5 is the background matrix correction factor due to lithogenic effects (takes into account possible lithological variability) (*Chen et al., 2007*; *Zhang et al., 2009*).

## Statistical analysis of results

Analytical data were checked for normality prior to statistical evaluation using the Kolmogorov–Smirnov (K-S) test. In case of acceptance, differences among concentrations of THg in water, sediment, fish and yam samples were determined using one-way analysis of variance (ANOVA). The Kruskal-Wallis test was applied when data did not follow a normal distribution and previous transformation did not normalize them. To identify the source of significant differences between groups, Tukey pairwise and Dunn's multiple comparison tests were done for parametric and non-parametric data respectively. These statistical analyses were performed at a 95% confidence interval (with differences in mean values accepted as being significant at $p < 0.05$) using Sigma Plot statistical software (v 14.0, Systat Software Inc., San Jose, CA, USA).

**Table 1  Mercury content of water, sediments, fish and yams from Namukombe stream.**

| Sample | Distance (m) | Total mercury concentration (mg/L or $\mu gg^{-1}$) | | | | | |
| --- | --- | --- | --- | --- | --- | --- | --- |
| | | Up sluice | | Middle sluice | | Down sluice | |
| | | Mean ± S.D | Range | Mean ± S.D | Range | Mean ± S.D | Range |
| Surface water | 0 | **1.21 ± 0.040** | 1.17–1.25 | **0.18 ± 0.070** | 0.11–0.25 | **0.10 ± 0.030** | 0.07-0.13 |
| | 10 | **0.15 ± 0.053** | 0.09–0.19 | **0.12 ± 0.017** | 0.11–0.14 | **0.08 ± 0.026** | 0.06-0.11 |
| | 20 | **0.12 ± 0.021** | 0.10–0.14 | **0.03 ± 0.026** | 0.01–0.06 | **0.02 ± 0.010** | 0.01-0.03 |
| | 30 | **0.09 ± 0.001** | 0.06–0.13 | **0.02 ± 0.010** | 0.01–0.03 | BDL | N/A |
| Superficial | 0 | 0.14 ± 0.040 | 0.10–0.18 | 0.11 ± 0.061 | 0.07–0.18 | 0.12 ± 0.010 | 0.11-0.13 |
| | 10 | 0.12 ± 0.035 | 0.10–0.16 | 0.02 ± 0.012 | 0.01–0.03 | 0.01 ± 0.001 | 0.009-0.011 |
| sediments | 20 | 0.03 ± 0.026 | 0.01–0.06 | 0.03 ± 0.021 | 0.02–0.05 | 0.02 ± 0.01 | 0.01-0.03 |
| | 30 | BDL | N/A | BDL | N/A | BDL | N/A |
| Fish (*Oreochromis nilotica*) | 0 | 0.11 ± 0.014 | 0.09–0.15 | 0.08 ± 0.010 | 0.07–0.09 | 0.08 ± 0.036 | 0.05-0.12 |
| | 10 | 0.04 ± 0.030 | 0.02–0.07 | 0.03 ± 0.035 | 0.01–0.07 | BDL | N/A |
| | 20 | BDL | N/A | BDL | N/A | BDL | N/A |
| | 30 | BDL | N/A | BDL | N/A | BDL | N/A |
| Yams (*Dioscorea alata*) | 0 | 0.30 ± 0.173 | 0.20–0.50 | 0.28 ± 0.026 | 0.26–0.31 | 0.29 ± 0.066 | 0.23-0.36 |
| | 10 | 0.24 ± 0.060 | 0.18–0.30 | 0.20 ± 0.030 | 0.17–0.23 | 0.15 ± 0.046 | 0.11-0.20 |
| | 20 | 0.12 ± 0.020 | 0.10–0.14 | 0.10 ± 0.01 | 0.06–0.15 | 0.01 ± 0.003 | 0.008-0.013 |
| | 30 | BDL | N/A | BDL | N/A | BDL | N/A |

**Notes.**

S.D, Standard deviation; BDL, below method detection limit of 0.001 mg/L or 0.001 $\mu gg^{-1}$; N/A, Not applicable.

Mean values in bold are higher than the maximum US EPA compliance limit of 0.002 mg/L for mercury in drinking water.

## RESULTS AND DISCUSSION

The mercury content of water, sediments, fish and yams from the different sluices of Namukombe stream with their descriptive statistics are given in Table 1.

### Water

Mean total mercury content of the water samples ranged from <detection limit to 1.21 ± 0.040 mg/L. Total mercury in water samples from up sluice were initially high at 0 m but reduced drastically after a distance of 10 m from the gold recovering site. There was a gradual decrease from 1.21 ± 0.040 to 0.09 ± 0.001 mg/L up sluice, 0.18 ± 0.070 to 0.02 ± 0.01 mg/L in middle sluice and 0.10 ± 0.030 mg/L until no detection down sluice (Table 1).

The observed decrement could be due to the fact that Namukombe stream is swampy, thus the flow of water is reasonably slow. Namukombe stream is rich in organic matter, thus some of the mercury might have been retained within the sediments. It is reported that organic matter can increase mercury methylation by stimulating heterotrophic bacteria (*Pseudomonas* species) in aerobic condition or by abiotic methylation (*International Programme on Chemical Safety, 1989*; *Ramesh et al., 2012*). In anaerobic condition, mercury react with organic carbon in the sediments to form toxic methyl and di-methyl mercury (*Ramesh et al., 2012*). Some anaerobic bacteria that possess methane synthetase are also reported to be capable of mercury methylation (*Wood & Wang, 1983*).

Further, owing to their static nature, sediments tend to get enriched with toxic materials more than water, which can undergo relatively rapid self-purification. Thus, a greater percentage of THg in an aquatic system is expected in sediments if there is effective binding with organic carbon bearing particles. This may innocuously retard the transfer of Hg to overlying water through interstitial water (*Gilmour, Henry & Mitchell, 1992*).

Heavy metals, which are less soluble in water such as Hg, are easily adsorbed and accumulated in sediments (*Alvarez et al., 2007*). However, in the event that the trace metal cannot be permanently adsorbed by sediments, it is released back to the overlying water when environmental conditions such as salinity, resuspension, pH, redox potential and organic matter decay rate changes (*Caille et al., 2003*; *Hill, Simpson & Johnston, 2013*). High (alkaline) pH and a decrease in chloride ion content, for example, enhances the mobility of mercury in aquatic environment, thereby favoring adsorption of mercury with oxides of iron, aluminium and silicon (*Gabriel & Williamson, 2004*) as well as clay. Abiotic inorganic mercury have also been reported to reduce in the presence of electron donors and low redox potentials (*Kim, Rytuba & Brown, 2004*). *Lino et al. (2019)* reported that suspended particles are the main carrier of mercury in water column and sediments. Further, biotic and abiotic factors seem to interact in a complex way in an aquatic ecosystem, causing marked variations in mercury concentrations in matrices (*Lino et al., 2019*). These explain the low levels of mercury reported in sediments than in water.

The levels of THg reported in water by this study is comparable to that of *Oladipo et al. (2013)* and *Mahre et al. (2007)* (Table 2). Further, mercury content of water in this study is concordant with preceding investigations which concluded that the quality of water in the periphery of mercury-based ASGM sites in Uganda have been deteriorated by mercury pollution (*Nabaasa, 2016*; *Mubende District Local Government, 2013*).

All the THg concentrations of the water samples except one (8.3%) sample from 30 m down sluice in this study were higher than the US EPA maximum contamination level of 0.002 mg/L for mercury in drinking water. Therefore, 91.7% of water sampled in this study is not safe for drinking.

## Superficial sediments

Sediments are good hosts of highly toxic pollutants from natural and anthropogenic sources (*Duyusen & Gorkem, 2008*) and have been reported as the biggest sink and major reservoir for heavy metals (*Yu et al., 2008*; *Orson, Simpson & Good, 1992*; *Singh, Huerta-Espino & William, 2005*; *Marchand et al., 2006*; *Ahmad et al., 2010*; *Bastami et al., 2012*; *Tavakoly Sany et al., 2013*; *Alves et al., 2014*). Sediments enhance accumulation of heavy metals in benthic invertebrates, thereby transferring them to higher levels of the food chain (*Sayadi et al., 2015*; *Shirneshan et al., 2013*; *Yap et al., 2002*; *Long et al., 1996*; *Fichet et al., 1999*). Therefore, monitoring sediments can give better understanding of trace metal contamination of aquatic ecosystems (*Gognou & Fisher, 1997*; *Birch, 2003*; *Morillo, Usero & Gracia, 2004*; *Aderinola et al., 2009*; *Abdollahi et al., 2013*; *Islam et al., 2015*; *Ali et al., 2016*) compared to water and/or floating aquatic plants which tend to give inaccurate estimations due to water discharge fluctuations and lower resident times.

**Table 2 Total mercury content of water, sediments, fish and yams in artisanal and small-scale gold mining areas reported by global studies.**

| Study area | Sample (matrix) | Total mercury content (mg/L or μgg) | Year[a] | Authors |
|---|---|---|---|---|
| Manyera River, Nigeria | Sediment | 0.018 | | |
| | Water | 0.021 | 2013 | Oladipo et al. |
| | Fish (*Heterotis niloticus*) | 0.008 | | |
| River Kaduna, Nigeria | Water | Range: 1.72–2.50 | 2007 | Mahre et al. |
| | Fisheries and aquatic life | Range: 0.0001–0.001 | | |
| Pra river basin, Ghana | Sediment | Average: 0.265 | 2006 | Donkor et al. |
| Rwamagasa artisanal gold mining area, Tanzania | Sediments | | | |
| | (a) Uvinza on the Malagarasi river | Range: 0.00017 to 0.00024 | | |
| | (b) Ilagala | Range: 0.10 to 0.66 | 2005 | Taylor et al. |
| | Yams | Range: 0.007 to 0.092 | | |
| Nambija, Ecuadorian Amazon | Sediment | Range: 0.7–9.3; background 0.5 | 2003 | Ramirez-Requelme et al. |
| Tongguan, Shaanxi Province, Peoples Republic of China | Sediment | Range: 0.3–0.9 | 2006 | Feng et al. |
| Buyat Bay, Indonesia | Sediment | Range: 0.010–0.017 | 2010 | Lasut et al. |
| Phichit Province, Thailand | Sediment | Range: 0.096–0.402 | 2007 | Pataranawata et al. |
| Tatelu gold mining area, Indonesia | Freshwater fish | 0.58 ± 0.44; greater than 45% of fish had total mercury above WHO maximum limit | 2006 | Castilhos et al. |
| Nilambur, Kerala, India | Sediment | Range: 0.103–0.468 | 2012 | Ramesh et al. |
| Namukombe stream, Busia gold district, Uganda | Water | Range: below detection limit to 1.21 | | |
| | Sediment | Range: below detection limit to 0.14 | | |
| | Fish (*Oreochromis nilotica*) | Range: below detection limit to 0.11 | 2019 | **Current study** |
| | Yams (*Dioscorea alata*) | Range: below detection limit to 0.30 | | |

**Notes.**
[a]Years reported represent the year the data were published, with most data collected 1–2 years prior to publication.

Mean THg concentrations of sediments from up sluice ranged from <detection limit to $0.14 \pm 0.040$ μgg$^{-1}$ while in the middle sluice, THg content of sediments ranged from <detection limit to $0.11 \pm 0.061$ μgg$^{-1}$. Down sluice, THg content of sediments ranged from <detection limit to $0.12 \pm 0.01$ μgg$^{-1}$ (Table 1). All the mean THg concentrations of the sediments from the three sluices were lower than the maximum permissible limit of 0.15 μgg$^{-1}$ recommended by US EPA 2001 standard (*US EPA, 2001*). For all the sluices, THg concentration in the sediments reduced with distance (Table 1) from the point source of pollution. No mercury was detected in sediments sampled 30 m away from all the gold recovery sites.

The mean THg content of the sediments in this study is lower than the values registered in other global studies such as that reported by *Donkor et al. (2006)*, *Ramirez-Requelme et al. (2003)* and *Feng et al. (2006)* (Table 2). However, *Ramesh et al. (2012)*, *Oladipo et al. (2013)*, *Lasut et al. (2010)*, *Pataranawata et al. (2007)* and *Taylor et al. (2005)* reported

lower THg content of sediments comparable to the THg concentrations recorded in this study (Table 2).

THg content of all the sediment samples except one i.e., sample from 0 m up sluice ($0.14 \pm 0.040$ $\mu gg^{-1}$) were below Threshold Effect Level (TEL) of 0.13 $\mu gg^{-1}$. All the THg concentrations of the sediments were lower than the Probable Effect Level (PEL) of 0.70 $\mu gg^{-1}$ for Hg in sediments postulated by *Smith et al. (1996)* and *MacDonald (1994)*. THg in the sediments lying between TEL and PEL is expected to be associated with adverse biological effects (*Ramesh et al., 2012*). Also, among the bottom sediment samples, none had mean THg higher than the background concentration of 0.25 $\mu gg^{-1}$, which is considered as normal in non-contaminated sediments (*Forstner & Wittman, 1981*). It is worth noting that the mercury content of the sediments in this stream were lower than the THg content of water from the corresponding sluices.

In this study, the retention rates of Hg in sediments, which is influenced by many factors such as the metallic forms of mercury (i.e., elemental, ionic, organic, or inorganic), pH, temperature, organic carbon and electrical conductivity were not investigated. Because the sediment Hg retention rates can vary from one location to another, the observed variability in THg concentrations in sediments from the sluices can be attributed to the differences in the sediment Hg retention rates, the distance from the source of the pollutant and the level of pollution from ASGM activities by the gold recovering sites. According to the Sediment Quality Criteria for Protection of Aquatic Life (Environment Canada, 1992 cited in *Haines et al., 1994*), all the sediments in this study had mean THg below the toxic threshold of 1.0 $\mu gg^{-1}$ and minimal effects threshold of 0.20 $\mu gg^{-1}$.

### *Oreochromis nilotica*

Fish from all the sluices had mean THg in the range of <detection limit to $0.11 \pm 0.014 \mu gg^{-1}$ (Table 1). The mean THg content of all the fish samples from the stream did not exceed the maximum WHO permissible limit for mercury in fish for human consumption (0.50 $\mu gg^{-1}$) as well as the WHO recommended limit for vulnerable groups (0.20 $\mu gg^{-1}$).

The highest mean THg content of fish reported in this study is lower than the mean THg reported by *Castilhos et al. (2006)* (Table 2). *Oladipo et al. (2013)* and *Mahre et al. (2007)* reported THg content of edible fish muscles lower than is reported in *O. nilotica* tissues by the current study (Table 2).

In this investigation, fish samples obtained from areas close to the gold recovery sites had higher THg content than those obtained from sampling points far from the gold recovery sites (Table 1). This could be due to a reduction in the mercury content of water as it spreads from the source of pollution. Mercury could have probably got entrained in the sediments as it spread outwards.

It is reported that fish ingest heavy metals by direct uptake in aqueous solution or by epithelial absorption of heavy metal contaminated water that sluices through their gills, skin, oral cavity and digestive tract (*Burger et al., 2002*). However, chronic intake of heavy metals by fish rest entirely on the metal concentration, volume of the ingested contaminated food or water, the heavy metal uptake speed, exposure duration, uptake route, ecological conditions external to the fish (including availability of water, temperature, pH) and innate

factors such as fish age (*Ahmed & Shuhaimi-Othman, 2010*), fish nutritional habits as well as the dynamic processes involved in the trace metal metabolism when ingested (*Koca et al., 2005*). Therefore, the lower levels of Hg recorded in this study could be because the fish samples were not so aged and the fact that *O. nilotica* is non-piscivorous. This is corroborated by the reports of *Mol et al. (2011)* who reported that the THg concentrations in freshwater piscivorous fish species in ASGM areas of Suriname, South America was 0.71 $\mu$gg$^{-1}$, about 3.7 times higher than 0.19 $\mu$gg$^{-1}$ recorded in non-piscivorous species in the same mercury-contaminated water bodies. *Marrugo-Negrete, Norberto & Olivero-Verbel (2008)* reported a similar finding in which carnivorous fish species: *Caquetaia kraussi*, *Hoplias malabaricus* and *Plagioscion surinamensis* in an aquatic system impacted by gold mining in Northern Colombia had high mean THg concentrations of 1.09 $\pm$0.17, 0.58 $\pm$ 0.05 and 0.53 $\pm$ 0.07 $\mu$gg$^{-1}$ fresh weight respectively while the lowest THg (0.157 $\pm$0.01 $\mu$gg$^{-1}$ fresh weight) was reported in a non-carnivorous fish species, *Prochilodus magdalenae*.

   *Weber et al. (2013)* reported that aquatic organisms exposed to copious levels of waterborne trace metals bioconcentrate the metals upon absorption, ultimately transferring them to humans as they are inevitable in human nutrition. Thus, for the general population, dietary intake is the dominant exposure pathway to mercury. Though the levels of THg in the edible muscles of *O. nilotica* consumed by the local community of Syanyonja village registered in this study are evidently low, the effect of mercury accumulation should not be overruled as other metabolically active organs such as the gills, liver and kidneys might contain higher THg concentrations (*Omara et al., 2018*). Dietary exposure to mercury has been proven to cause elevated risks of cardiovascular diseases with severe exposure negatively impacting the reproductive and immune systems (*Rae & Graham, 2012*; *AMAP, 2003*).

### *Dioscorea alata*
The mean THg content of yams (*D. alata*) from Namukombe stream varied between <detection limit to 0.30 $\pm$ 0.173 $\mu$gg$^{-1}$ (Table 1). Yams from up sluice (at 0 m) had the highest mean THg of 0.30 $\pm$ 0.173 $\mu$gg$^{-1}$. Middle sluice samples at 0 m had THg content of 0.28 $\pm$ 0.026 $\mu$gg$^{-1}$, while down sluice samples at 0 m had the highest mean THg content of 0.29 $\pm$ 0.066 $\mu$gg$^{-1}$ (Table 1). There was no appreciable difference in the THg content of the yams from the different sluices. This trend can be related to the levels of Hg in both water and sediments from the sluices in relation to the distance of the samples from the ASGM activities. The highest mean THg content of yams grown in Namukombe stream is quite higher than that reported by *Taylor et al. (2005)* (Table 2). It is worth noting that yams in this stream recorded the highest THg (0.30 $\pm$ 0.173 $\mu$gg$^{-1}$) of all the studied matrices. This could be because yams are exposed to diverse routes of contamination by mercury from ASGM activities such as the sediments (soils), contaminated water and atmospheric disposition on leaves during growth. However, the mean THg content of yams reported could probably be higher than recorded due to losses during ashing at 550 °C for 5 h.

Overall, the results of this study are concordant with other global studies which observed elevated THg concentrations in water and sediments, with patterns of decreasing THg concentration with distance from ASGM sources (*Gray et al., 2002*; *Nartey et al., 2011*; *Diringer et al., 2015*).

## Health risk assessment from consumption of fish and yams and skin contact with sediments from Namukombe stream

Chronic low level intake of priority trace metals such as mercury have been implicated for deleterious human health effects, which become apparent following years of persistent exposure (*Bortey-Sam et al., 2005*; *Huang et al., 2007*; *Liu, Probst & Liao, 2005*). Target hazard quotient method was used to assess the potential health risks of Hg accumulation through consumption of the edible muscles of *O. nilotica* and *D. alata* as well as dermal contact during ASGM.

The estimated daily intakes (EDIs) ranged from 0.0049 to 0.0183 $\mu gg^{-1}day^{-1}$ and 0.020 to 0.073 $\mu gg^{-1}day^{-1}$ for fish consumed by adults and children respectively. The corresponding health risk indices (HRIs) ranged from 0.0123 to 0.0458 and 0.05 to 0.183 (Tables 3 and 4). In 1960s, Minamata residents of Japan suffered unprecedented neuropathies due to consumption of mercury-contaminated seafood (*Japan Environmental Agency, 1989*). More so, some studies have reported that mercury is toxic to fish (*Tilapia guineersis* and *Tilapia fuscatus*) and induce fish weight loss even on exposure to sub-lethal doses of mercury in more than two fortnights (*Oyewo & Don-Pedro, 2003*). Thus, *O. nilotica* in Namukombe stream is endangered due to the detected mercury pollution of the fish.

The EDIs were from 0.0042 to 0.1279 $\mu gg^{-1}day^{-1}$ and 0.013 to 0.394 $\mu gg^{-1}day^{-1}$ for yams consumed by adults and children respectively. The statistical HRIs recorded were from 0.011 to 0.320 and 0.033 to 0.985 respectively (Tables 3 and 4). The HRI of 0.985 registered for consumption of yams from 0 m up sluice by children is very close to 1.0, implying that consumption of yams from this site by children might lead to mercury-related health risks.

The $ADD_{therm}$ ranged from $1.015 \times 10^{-6}$ to $7.105 \times 10^{-6} \mu gg^{-1}day^{-1}$ and $7.47 \times 10^{-7}$ to $5.227 \times 10^{-6} \mu gg^{-1}day^{-1}$ (Tables 3 and 4) for contact with mercury-contaminated dredged sediments from Namukombe stream by adults and children respectively. The HRIs respectively ranged from $1.015 \times 10^{-4}$ to $7.105 \times 10^{-4}$ and $7.47 \times 10^{-5}$ to $5.227 \times 10^{-4}$ for adults and children (Tables 3 and 4).

Target hazard quotient of less than unity (1.0) indicate the relative absence of health risks associated with intake of Hg through consumption of either mercury contaminated fish, yams or contact with sediments. However, ingestion of both fish and yams, coupled with persistent dermal exposure to mercury in sediments during panning would lead to potential health risks especially for children.

## Mercuric accumulation based on bioaccumulation factors

The bioaccumulation factors: bioconcentration factor and biota to sediment accumulation factor computed for *O. nilotica* in Namukombe stream are presented in Table 5. The

Omara et al. (2019), *PeerJ*, DOI 10.7717/peerj.7919

**Table 3  Toxicity indices of mercury from consumption of fish and yams and contact with sediments from Namukombe stream by adults.**

| Distance (m) | Fish (*Oreochromis nilotica* Lin.) | | | | | | Yams (*Dioscorea alata*) | | | | | | Superficial sediments | | | | | |
|---|---|---|---|---|---|---|---|---|---|---|---|---|---|---|---|---|---|---|
| | EDI (µg/kg/day) | | | THQ | | | EDI (µg/kg/day) | | | THQ | | | ADD $_{therm}$ (µg/kg/day) $\times 10^{-6}$ | | | THQ $\times 10^{-4}$ | | |
| | Up sluice | Middle sluice | Down Sluice | Up sluice | Middle sluice | Down sluice | Up sluice | Middle sluice | Down sluice | Up sluice | Middle sluice | Down sluice | Up sluice | Middle sluice | Down sluice | Up sluice | Middle sluice | Down sluice |
| 0 | 0.0183 | 0.0133 | 0.0133 | 0.0458 | 0.0333 | 0.0333 | 0.1279 | 0.1194 | 0.1237 | 0.320 | 0.299 | 0.309 | 7.105 | 5.583 | 6.090 | 7.105 | 5.583 | 6.090 |
| 10 | 0.0067 | 0.0049 | N/A | 0.0168 | 0.0123 | N/A | 0.1023 | 0.0853 | 0.0639 | 0.256 | 0.213 | 0.160 | 6.090 | 1.015 | 0.5075 | 6.090 | 1.015 | 0.5075 |
| 20 | N/A | N/A | N/A | N/A | N/A | N/A | 0.0512 | 0.0426 | 0.0042 | 0.128 | 0.105 | 0.011 | 1.523 | 1.523 | 1.015 | 1.523 | 1.523 | 1.015 |
| 30 | N/A | N/A | N/A | N/A | N/A | N/A | N/A | N/A | N/A | N/A | N/A | N/A | N/A | N/A | N/A | N/A | N/A | N/A |

**Notes.**

EDI, Estimated daily intake; THQ, Target Hazzard Quotient (unitless); ADD, average daily dose; N/A, Not applicable.

Omara et al. (2019), *PeerJ*, DOI 10.7717/peerj.7919

**Table 4  Toxicity indices of mercury from consumption of fish and yams and contact with sediments from Namukombe stream by children.**

| Distance (m) | Fish (*Oreochromis nilotica* Lin.) | | | | | | Yams (*Dioscorea alata*) | | | | | | Superficial sediments | | | | | |
|---|---|---|---|---|---|---|---|---|---|---|---|---|---|---|---|---|---|---|
| | EDI (μg/kg/day) | | | THQ | | | EDI (μg/kg/day) | | | THQ | | | ADD therm (μg/kg/day) $\times 10^{-6}$ | | | THQ $\times 10^{-4}$ | | |
| | Up sluice | Middle sluice | Down sluice | Up sluice | Middle sluice | Down sluice | Up sluice | Middle sluice | Down sluice | Up sluice | Middle sluice | Down sluice | Up sluice | Middle sluice | Down sluice | Up sluice | Middle sluice | Down sluice |
| 0 | 0.073 | 0.053 | 0.053 | 0.183 | 0.133 | 0.133 | 0.394 | 0.367 | 0.381 | 0.985 | 0.918 | 0.953 | 5.227 | 4.107 | 4.480 | 5.227 | 4.107 | 4.480 |
| 10 | 0.027 | 0.020 | N/A | 0.0665 | 0.05 | N/A | 0.315 | 0.262 | 0.197 | 0.788 | 0.655 | 0.493 | 4.480 | 0.747 | 0.373 | 4.480 | 0.747 | 0.373 |
| 20 | N/A | N/A | N/A | N/A | N/A | N/A | 0.158 | 0.131 | 0.013 | 0.395 | 0.328 | 0.033 | 1.120 | 1.120 | 0.747 | 1.120 | 1.120 | 0.747 |
| 30 | N/A | N/A | N/A | N/A | N/A | N/A | N/A | N/A | N/A | N/A | N/A | N/A | N/A | N/A | N/A | N/A | N/A | N/A |

**Notes.**

EDI, Estimated daily intake; THQ, Target Hazzard Quotient (unitless); ADD, average daily dose; N/A, Not applicable.
**Table 5 Bioconcentration factor, Biota to Sediment Accumulation Factor, Contamination factor and Geoaccumulation Index for the investigated matrices from Namukombe stream.**

| Distance (m) | Bioconcentration factor | | | Biota to sediment accumulation factor | | | Contamination factor | | | Geoaccumulation index | | |
|---|---|---|---|---|---|---|---|---|---|---|---|---|
| | Up sluice | Middle sluice | Down sluice | Up sluice | Middle sluice | Down sluice | Up sluice | Middle sluice | Down sluice | Up sluice | Middle sluice | Down sluice |
| 0 | 0.091 | 0.444 | 0.800 | 0.786 | 0.727 | 0.667 | 0.56 | 0.44 | 0.48 | −1.423 | −1.771 | −1.644 |
| 10 | 0.267 | 0.250 | N/A | 0.333 | 1.500 | N/A | 0.48 | 0.08 | 0.04 | −1.644 | −4.230 | −5.233 |
| 20 | N/A | N/A | N/A | N/A | N/A | N/A | 0.12 | 0.12 | 0.08 | −3.644 | −3.644 | −4.230 |
| 30 | N/A | N/A | N/A | N/A | N/A | N/A | N/A | N/A | N/A | N/A | N/A | N/A |

Notes.
N/A, Not applicable.

results show that there are higher mercury levels in *O. nilotica* tissues than in the surface water samples. Bioconcentration factors for mercury in *O. nilotica* were ranked as follows: down sluice >middle sluice >up sluice. The highest bioconcentration factor of 0.800 was recorded at 0 m down sluice while the lowest bioconcentration factor of 0.250 was recorded at 10 m middle sluice. Such trace metal accumulation levels in fish as in this concerted study augment published data reported by other authors on different species of aquatic organisms (*Benson et al., 2017*; *Avelar et al., 2000*; *Kwok et al., 2005*; *Zhao et al., 2012*). Therefore, this study suggests that *O. nilotica* is a sentinel organism for biomonitoring of aquatic ecosystems.

Biota to sediment accumulation factor explored the rate of Hg uptake from sediments and its subsequent accumulation in *O. nilotica* tissues. In this investigation, the highest biota to sediment accumulation factor of 1.500 was recorded at 10 m in the middle sluice while the lowest biota to sediment accumulation factor (0.333) was recorded at 10 m up sluice. Thus, mercury enrichment was highest in the middle sluice, though the sediments have higher concentrations of Hg than the edible muscles of *O. nilotica*.

## Quality of superficial sediments from Namukombe stream
### Contamination factor
All the statistical contamination factors were less than 1.0 (the highest statistical value of 0.56 was recorded at 0 m up sluice and the lowest value of 0.04 was reported at 10 m down sluice) (Table 5). According to *Hakanson (1980)*, four (4) contamination categories are distinguished: <1: low contamination; $1 \leq$ factor <3: moderate contamination, $3 \leq$ factor <6: considerable contamination and factor >6: very high contamination. Thus, basing on the criteria, there is very low contamination of sediments in the studied parts of Namukombe stream.

## Geoaccumulation index
Müller geoaccumulation index ($I_{geo}$) is a frequently employed analytical index for examination of the contamination level of sediment samples by trace metals. It assesses the degree of contamination by comparing the current levels of trace metal concentrations to the previous status of the research site. The computed Müller geoaccumulation indices for the bottom sediments from Namukombe stream ranged from −5.233 to −1.423 (Table 5).

The $I_{geo}$ is composed of seven grades along with associated sediment quality levels according to the degree of trace metal pollution. The values are classified as follows: no contamination ($I_{geo} < 0$); low to median contamination ($I_{geo}$ between 0 and 1); median contamination ($I_{geo}$ between 1 and 2); median to strong contamination ($I_{geo}$ between 2 and 3); serious contamination ($I_{geo}$ between 3 and 4); serious to extreme contamination ($I_{geo}$ between 4 and 5); and extreme contamination ($I_{geo} > 5$).

In this study, the geoaccumulation indices were all negative for the sluices (Table 5), reflecting that there is no serious anthropogenic pollution of the studied sluices in Namukombe stream.

## Conclusion and recommendations

Persistent utilization of mercury in ASGM in Syanyonja and the proliferation of its environmental and human health effects pose significant challenges to sustainability. Water in Namukombe stream is contaminated with up to $1.21 \pm 0.070$ mg/L of Hg which is above US EPA maximum permissible limit for Hg in drinking water. The maximum THg content of sediments from the stream is $0.14 \pm 0.040\ \mu gg^{-1}$ which is lower than the maximum limit of $0.150\ \mu gg^{-1}$ recommended by US EPA 2001 standard. Release of ASGM residual Hg into Namukombe stream have resulted in significant entrainment of Hg in water and sediments in the stream. The mercuric content of the edible whole muscles of the locally consumed fish (*O. nilotica*) is lower than in sediments, yams and drinking water.

THg content of the edible whole muscles of fish from Namukombe stream ranges from <detection limit to $0.11 \pm 0.010\ \mu gg^{-1}$ which is still within the maximum WHO permissible limit of $0.5\ \mu gg^{-1}$ for Hg in fish for human consumption. Health risk assessment indicate that consumption of *D. alata* from 0 m up sluice may have a potential health risk, particularly to children.

Further research should determine the geochemical properties (pH, organic carbon and conductivity) of the sediments as these properties tend to correlate with limnologic mercury accumulation in sediments. Research should be done to evaluate the mercury content of metabolically active organs (gills, liver, kidney) of *O. nilotica*. The levels of methyl mercury and other trace metals such as Lead and Arsenic should be determined in water, sediments, yams, fish as well as soils. The atmospheric flux of mercury in the study area should be determined.

# ACKNOWLEDGEMENTS

The technical advices of Joseph Ddumba Lwanyaga and Marion Engole of Busitema University and Robert Gazetti of Uganda Industrial Research Institute, Nakawa are appreciated.

## Funding
The authors received no funding for this work.

## Competing Interests

Timothy Omara works at AgroWays Uganda Limited. Raymond Kalukusu, Brenda Victoria Nakabuye and Bashir Musau are employed by Leading Distillers Uganda Limited. Sarah Kagoya is employed by Kakira Sugar Limited.

## Author Contributions

- Timothy Omara conceived and designed the experiments, performed the experiments, contributed reagents/materials/analysis tools, authored or reviewed drafts of the paper, approved the final draft, literature search.
- Shakilah Karungi conceived and designed the experiments, performed the experiments, analyzed the data, contributed reagents/materials/analysis tools, prepared figures and/or tables, authored or reviewed drafts of the paper, approved the final draft.
- Raymond Kalukusu and Brenda Victoria Nakabuye analyzed the data, prepared figures and/or tables, authored or reviewed drafts of the paper, approved the final draft.
- Sarah Kagoya analyzed the data, contributed reagents/materials/analysis tools, prepared figures and/or tables, authored or reviewed drafts of the paper, approved the final draft, grammatical proofreading.
- Bashir Musau performed the experiments, contributed reagents/materials/analysis tools, authored or reviewed drafts of the paper, approved the final draft.

## Animal Ethics

The following information was supplied relating to ethical approvals (i.e., approving body and any reference numbers):

Approval to carry out research in this study area was informally granted by the local council 1 of the village upon request using an introduction letter from Department of Mining and Water Resources Engineering, Faculty of Engineering, Busitema University, Tororo, Busitema, Uganda. The introduction letter was given after a proposal indicating how the principal investigator (Shakilah Karungi) will ensure safe use and disposal of the toxic chemicals for mercury analysis in the tissues and water, sediments, fish carcasses and yams was submitted to the Department committee.

## Field Study Permissions

The following information was supplied relating to field study approvals (i.e., approving body and any reference numbers):

Field experiments were approved by Department of Mining and Water Resources Engineering, Faculty of Engineering, Busitema University, Tororo, Busitema, Uganda for Shakilah Karungi (Approval No. BU/UG/2013/95).

## Data Availability

All the raw measurements are available in Table S2.

## Supplemental Information

Supplemental information for this article can be found online at http://dx.doi.org/10.7717/peerj.7919#supplemental-information.

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
