# Peer review of "Mercuric pollution of surface water, superficial sediments, Nile tilapia (Oreochromis nilotica Linnaeus 1758 [Cichlidae]) and yams (Dioscorea alata) in auriferous areas of Namukombe stream, Syanyonja, Busia, Uganda"

_PeerJ, doi:10.7717/peerj.7919_

## Round 0.1 · original submission · Major Revisions

We have received the reviews of your paper, and, although one of the reviewers recommended a reject, I believe the authors are able to carry out all suggestions and revisions necessary in order to make the draft publishable in PeerJ. Especially, comparisons to the literature should be carried out using the same concentration units, please take care with this. In addition, the methodological section requires extensive detailing, such as recovery CRM values which are paramount. Also, English MUST be reviewed by a native English speaker. As is, it is not adequate for publication. I hope to receive a revised manuscript soon that takes into account the reviewers comments and suggestions.

·

Basic reporting

The context of fish and yam selection were not mentioned - was it because of locals diet ?

Experimental design

Author did not mentioned the following
1. What are the percentage of recovery for target metals in this experiment?
2. Any information on CRM?
3. How does the author perform the QA/QC for this experiment?

Validity of the findings

No comment.

Additional comments

Line 63 – What is ASGM and how dies it differs from ASM?

Line 51 – ‘Other safe gold recovery methods” – how do you sure the other method is safer?

Line 290 = How do you explain the high levels of THg recorded in water compared to sediments for the corresponding sluice in relation to environmental conditions such as salinity, resuspension, pH, redox potential etc?

The author mentioned that fish samples were not so aged – how does this relate to people consuming of fish? Could be that their are not consuming this fish due to the small size?

Why the author study yam? What are the significant?

Line 147 – Were the fish samples are of edible size to be consumed by locals?

Why choose this species of fish? What is their relation to the human health?

Line 170 – Please put these info into table for easy reference

Line 268, 314 Mean ± standard deviations decimal points consistency

Line 274 – Please define 'slow flow' in your text – have you measured the flow rate?

Line 322 – Authors should put all these info into a table for an easy reference

Line 363 – Why the level differs since fish is a mobile organism and would not be staying in the same area. Are those study area separated and prevent migration of fish ?

Line 357 – Should put all these info into a table for easy reference

Line 374 - Fish samples were not so aged; - then does this reflect the fish been consumed by locals?

Line 375 – Why select this species of non-piscivorous fish ‘O. nilotica is non-piscivorous.’?

Line 383 = “95% of Hg in most fish exists as MeHg”
Then why are you not measuring MeHg?

Line 420 – the sentence containing ‘Thus, it can be deduced that O. 420 nilotica in Namukombe stream is endangered.’ is a hanging statement.
Author must explain endangered due to what, in a single sentence.

Line 481 – “water in Namukombe stream is contaminated with up to 1.21±0.070mg/L of Hg which is above US EPA maximum permissible limit for Hg in drinking water.”

Why are you comparing raw versus treated water?

Figure 1 – no demarcation on the location/position of the river and mining area found.
Explain your Recovery Percentage, What is your CRM and how do you conduct the QA/QC for this research.

·

Basic reporting

This study reports an interesting study involving Hg pollution in an ecologically sensitive area of Uganda. Sufficient literature have been consulted and cited and also good number of environmental samples were collected and studied for Hg which is a toxic heavy metal. Introduction is well written and justifies the purpose of the study.

However, the manuscript is not written in good English. For example, L353- All the fishes from the stream did not exceed the maximum WHO….. Should be better to write as- Any fish sample from the stream did not ……

In some instances basic writing etiquette not followed Line 63: Expand ASGM at a first instance of mention. Is this synonymous with ASM mentioned in lines 57 and 58? If so, change ASM to ASGM such that abbreviations are used consistently to avoid any confusion. Authors seem to have used yam species D. alata for their study. However they have used the text ‘yams’ right from the title. It is not clear and mentioned what the other species of yam used was? If only one species used reword ‘yams’ throughout the manuscript.

There is only one figure (map) presented. This figure is not descriptive and labelled. For example, authors could have indicated sluice locations on the map.

Raw data of lab analysis is not shared.

Experimental design

Article fits into the aims and scope the journal. Research question is well defined and relevant but study design have flaws or not clearly explained. Methodology section is not explicitly and clearly described sampling site and sampling method and how sampling stations were determined. Sufficient description is lacking to repeat or replicate the study elsewhere. Authors should indicate and label up sluices, mid sluices etc in Fig 1. Otherwise, readers cannot understand relative location of the sampling stations. Also the sampling stations of yam material should be clear and well described. For example, I cannot comprehend how did authors found a yam sample at exactly ‘0 m’ from a sluice? Does ASGM activities carried out in a food garden or crop field? If so, where did the water samples come from? Besides, how did you confirm that Hg contamination/ pollution are not from a natural source? Authors should have taken few samples before the ASGM area to confirm. Without these samples from uncontaminated areas, this study findings are just status reporting and cannot pinpoint THg found is due to the ASGM activities.

Validity of the findings

Regarding analysis of data-Have you checked for the normal distribution of the data? Generally, the data from such studies won’t show normal distribution. As a consequence, the parametric test that was used is not appropriate for such data. As raw data is not provided could not be verified. What type of standards was used and what type of quality check measures were undertaken in lab analysis? No mention of quality checks in the manuscript. There should be a brief explanation. Conclusions are well stated and no issues.

Reviewer 3 ·

Basic reporting

The paper deal with an interesting issue that is few investigated in the studied site. I think it is an important subject that will be of interest to the readers of the journal. However, I suggest improving the scientific writing to better understand the scientific problem (putting in a global perspective). Some grammatical errors need to be fixed.

Suggestion:
- Lines 57-86: The problem with the use of Hg in gold mining is worldwide. I suggest putting this problem in a worldwide perspective, indicating that this same problem occurs in many sites. After that, the text can conduct the reader into the study area. Currently, the paper is very regional.
Many papers can be used for this purpose. For example:
Chibunda, R.T., Pereka, A.E., Tungaraza, C., 2008. Effects of sediment contamination by artisanal gold mining on Chironomus riparius in Mabubi River, Tanzania. Phys. Chem. Earth 33, 738-743. https://doi.org/10.1016/j.pce.2008.06.051.
Donkor, A.K., Bonzongo, J.C., Nartey, V.K., Adotey, D.K., 2006. Mercury in different environmental compartments of the pra river basin, Ghana. Sci. Total Environ. 368, 164-176. https://doi.org/10.1016/j.scitotenv.2005.09.046.
Feng, X., Dai, Q., Qiu, G., Li, G., He, L., Wang, D., 2006. Gold mining related mercury contamination in Tongguan, Shaanxi Province, PR China. Appl. Geochem. 21, 1955-1968. https://doi.org/10.1016/j.apgeochem.2006.08.014.
Gerson, J.R., Driscoll, C.T., Hsu-Kim, H., Bernhardt, E.S., 2018. Senegalese artisanal gold mining leads to elevated total mercury and methylmercury concentrations in soils, sediments, and rivers. Elem Sci Anth 6 (11), 1-14. http://doi.org/10. 1525/elementa.274.
Goix, S., Maurice, L., Laffont, L., Rinaldo, R., Lagane, C., Chmeleff, J., Menges, J., Heimburger, L., Maury-Brachet, R., Sonke, J.E., 2019. Quantifying the impacts of artisanal gold mining on a tropical river system using mercury isotopes. Chemosphere 219, 684-694. https://doi.org/10.1016/j.chemosphere.2018.12.036.
Limbong, D., Kumampung, J., Ayhuan, D., Arai, T., Miyazaki, N., 2005. Mercury pollution related to artisanal gold mining in north Sulawesi Island, Indonesia. Bull. Environ. Contam. Toxicol. 75, 989-996. https://doi.org/10.1007/s00128- 005-0847-0.
Lino, A.S., Kasper, D., Guida, Y.S., Thomaz, J.R., Malm, O., 2019. Total and methyl mercury distribution in water, sediment, plankton and fish along the Tapajós River basin in the Brazilian Amazon. Chemosphere, 235, 690-700. https://doi.org/10.1016/j.chemosphere.2019.06.212
Marrugo-Negrete, J., Benitez, L.N., Olivero-Verbel, J., 2008. Distribution of mercury in several environmental compartments in an aquatic ecosystem impacted by gold mining in Northern Colombia. Arch. Environ. Contam. Toxicol. 55 (2), 305-316. https://doi.org/10.1007/s00244-007-9129-7.
Miserendino, R.A., Guimaraes, J.R.D., Schudel, G., Ghosh, S., Godoy, J.M., Silbergeld, E.K., Lees, P.S.J., Bergquist, B.A., 2018. Mercury pollution in Amapá, Brazil: mercury amalgamation in artisanal and small-scale gold mining or land- cover and land-use changes? ACS Earth Space Chem 2 (5), 441-450. https://doi. org/10.1021/acsearthspacechem.7b00089.

Experimental design

The sampling design need to be better described. It is impossible understand exactly what was done. What do upstream, middle stream and downstream mean? Distance (0, 10, 20 and 30) from what?

I pointed some suggestions:
- Line 133: What kind of plastic was used?
- Lines 133-138 and 153-159: The method blank and its THg results need to be detailed (perhaps in sup. material) – example: bottle blank, equipment blank, blank of the filtration and analytical blank.
- Line 136: What kind of filter was used?
- Line 136: How much acid was used to preserve samples?
- Line 138: Were water and wasterwater assessed? Only the procedure of water collection was described.
- Line 142: What kind of plastic was used?
- Line 148: This section can present a brief analytical method of fish samples.
- Line 150: Some Hg species can be lost after 5 h at 550 C. The discussion needs to consider that.
- Lines 153-155: Sediment, fish and yam: Were these samples oxidized with BrCl? How much BrCl was used? How much time? At room temperature? The description of analytical method needs to be improved.
- Line 152: How was accuracy assessed?
- Lines 216-218: How were EF, PLI, RI and HG calculated? Whats about their results?
- Lines 230-231: How was the background defined?
- Lines 242-249: They are not results. They are method.
- Lines 244-248: How were heterocedasticity and normality evaluated? Table 1 shows spatial comparison (between 0, 10, 20 and 30 sites) – this statiscal analysis is not described here.
- I suggest adding the detection limit of the methods.

Validity of the findings

- The discussion seems consistent with the results, but a deeper analysis could not be done. It is not possible to evaluate the general context of the discussion because it is not possible to understand the experimental design.

- The results need to be better described. Na forma atual, os resultados são somente a apresentação de tabelas.
Uma sugestão: Lines 268-271, 313-315, 317-320, 352 and 394-398: this is result (not discussion). Nao tem 0 (abaixo do limite de detecção). O que é variou de tanto a tanto com desvio? Não dá para entender os valores.

- The results need to be better described. In the current format, the results are just the presentation of tables.
Suggestions:
Lines 268-271, 313-315, 317-320, 352 and 394-398: this is result (not discussion).
There isn't 0 (results are below detection limit).
It is difficult to understand the values. Are they mean? Min-Max?

- Conclusions and recommendations: It can be summarized.
Suggestions:
You can remove results.
In further study – I suggest putting suggestions of studies that can fill gaps of the evaluated processes or about new important questions in the area of study. For example: "evaluate the mercuric content of the different organs": I understand the importance of this. However, why is it important for gaps raised in the study? "Lead and arsenic": why? Are they released in the mining activities studied here?

- ADD: There are many chemical species of mercury in sediment samples. The contact with polluted sediment will not necessarily transfer the mercury to person. This needs to be incorporated into the discussion. The present study assessed only THg. It is impossible to know if Hg is in a chemical species that will be absorbed by the dermal contact.

Additional comments

Title
- It can be shortened to be simple and straightforward.
Example:
The words “surface”, “superficial” and “(Dioscorea alata)” can be removed.
“Nile tilapia (Oreochromis nilotica Linnaeus 1758 [Cichlidae])” can be replaced by “fish”.

Abstract
- Line 29: "pollution and contamination". It was not clear what the difference between the two. I suggest putting only one.
- Lines 34-35: I suggest adding the sample size of each sample type (e.g., water, fish).
- There is no 0. I suggest presenting the results "<detection limit".
- It is difficult to understand the results:
THg in water from 0 to 1.21. Min-max? Mean? Is 0.04 the standard deviation?
The same occurs for fish and yams.
Sediments: is 0.14 the mean? The maximum value?
Please, clarify what the values of the results mean.
- Line 46: 91.7% is not "some".

Introduction
- Line 63: What does ASGM mean? The acronym has not been defined previously.
- Lines 71-72: This statement needs a reference.
- Line 96: The mercury released by goldminig supplies the ecosystem with anthropogenic mercury. The mining activity does not transmogrify mercury.
- Line 113: I suggest using "sediment" instead of "sediments".
- Lines 112-116: It could be divided into three sentences because the objective of the study has 3 different ideas. Each could be presented in one sentence.

Discussion
Lines 321-322: “other global studies”: Are they in mining sites? Are they in natural sites? What is the quality of environment assessed in those studies? It is important to understand if they are comparable with the studied site.
Lines 337-338: Sites worldwide vary naturally in THg concentrations in sediments. A value considered as normal in non-contaminated sediments can not be applied to anywhere in the world. It has to be used very cautiously and is not suitable for a wide range of naturally enriched environments.
Lines 351, 438, 440 and other: "Linnaeus 1758 (Cichlidae)" may be cited only once (when the species name is first cited in the text-line 210). Throughout the paper, "Linnaeus 1758 (Cichlidae)" need not be written. I suggest writing only the species name.
Lines 356-362 and Table 1: The mercury concentration in fish, sediment and yam are more or less low. The cited studies also have low concentrations. I suggest to report the results of the present study and those of literature in ug.kg-1 instead of ug.g-1 to facilitate the visualization of the values.
Lines 367-368: I suggest "epithelial absorption" instead of "epithelial ingestion".

Figure 1
The figure needs to be improved. Examples:
I suggest putting a letter in each square and saying in the legend what they mean.
The three maps need to contain geographic coordinates, scale and the compass rose.
What do the symbols on the map mean? (e.g., lines, purple x)
Do the names refer to the states? Limits of basin drainage?

Tables
The caption needs to be improved. A caption must be self-explanatory. All acronyms in the table need to be explained in the caption or in the footnote.

Table 1
e.g.
SE- standard error?
SD – standard deviation?

Table 2 and 3
e.g.
EDI?
THQ?
If there is no result of 30, do not put this line. Why is there no result of 30?

Table 4
e.g.
ADD?
THQ?

The same for other tables.

General:
- Always separate unit of measurement from the number: e.g.: 30 m, 0.04 μg. Only % needs to be without space (e.g., 90%).
- There are many acronyms in the paper and this confuses the reader. The authors could evaluate which acronyms are really needed in the paper (e.g., frequently cited expressions) and which ones could be removed.

---

## Round 0.2 · Minor Revisions

Please take into account one of the reviewers' comments regarding distance from 0-10 m of yam sampling, and respond as required.

·

Basic reporting

Authors have now extensively revised the manuscript and have addressed most of my concerns. I see a large improvement in the clarity of the manuscript. Authors have added some nice figures. One concern I have in this version is use of the term "sluicing". For instance, sluicing referred by authors in Fig 2a is actually 'panning' and 'not sluicing' in this part of the word. I suggest authors to revisit and recheck appropriateness of the word they used and if possible to define and explain this a bit in introduction section.

Experimental design

Fine and understandable now. The concern is the one i raised on the earlier version. How did authors sampled yams at a distance of 0 m up sluice of the stream? This is not clarified and in the abstract it is written still as "Consumption of D. alata grown at 0 m up sluice of Namukombe stream may pose deleterious health risks as reflected by the health risk index of 0.985 being very close to one". In the revised version with a new figure (as Figure 4), and the ground situation as shown in the Fig 2a does not match the statement above. I think authors have taken yam samples within 0-10 m from sluicing location. If this is the case, should be explained properly.

Validity of the findings

Statistical methods are now clearly explained.

Additional comments

Revised well.

Reviewer 3 ·

Basic reporting

No comment.

Experimental design

No comment.

Validity of the findings

No comment.

Additional comments

I think the authors did a good job. The new draft presented resolutions for the topics raised or they were properly answered. I recommend this new version for publication.

---

## Round 0.3 · accepted · Accept

Dear authors, thank you for submitting a revised version of your manuscript taking into account the sampling distance, as was required. Your manuscript is now acceptable for publication in PeerJ.